# Assessing the Risks and Cultural Relativity of Diabetes in Black Individuals of African Caribbean Ancestry (ACB) Aged 18–39 Years in Toronto

**DOI:** 10.3390/ijerph22010085

**Published:** 2025-01-10

**Authors:** Akm Alamgir, Rhea Raghunauth, Osezua Momoh, Cliff Ledwos

**Affiliations:** 1Organizational Knowledge and Learning, Access Alliance Multicultural Health and Community Services, 340 College Street, Suite 500, Toronto, ON M5T 3A9, Canada; 2St. George Campus, University of Toronto, Toronto, ON M5S 1A1, Canada

**Keywords:** CANRISK, diabetes, African Caribbean Black, youth and young adults, HbA1c

## Abstract

**Context:** Diabetes rates are high in Black and some other ethnic communities, often leading to more severe complications. We conducted a study to identify the prevalence and risk of diabetes among African Caribbean Black (ACB) individuals aged 18–39 and to assess the sensitivity of glycated hemoglobin (HbA1c) compared to an oral glucose tolerance test (OGTT) to diagnose diabetes. **Methods:** In this mixed-methods study, maximum variation sampling was used to recruit 272 ACB participants from fourteen African and five Caribbean countries from Toronto. Participants’ height, weight, waist circumference, HbA1c, OGTT, demographic, and behavioural data were collected. SPSS was used to analyze the quantitative data. This study used descriptive statistics for frequency distribution and cross-tabulation while inferential statistics (regression, ANOVA, factor analysis, etc.) were used for relational analysis. Because of the small sample size, qualitative data were analyzed manually using the charting technique. **Results:** This study found that 1.5% of participants had diabetes, 9.2% had prediabetes, and 44.9% were at risk of developing diabetes. The mean value of HbA1c, FBS, and 2hPG was 5.5%, 4.8 mmol/L, and 5.7 mmol/L, respectively. The mean BMI was 28.2 kg/m^2^, and the waist circumference was 85.8 cm. This study found a correlation between glucose intolerance and increasing body mass index (BMI) and waist circumference (WC). Dietary habits, physical inactivity, and mental health challenges were risk factors among the participants. HbA1c was found to be a more sensitive and culturally acceptable screening measure than OGTT in diagnosing diabetes. **Conclusions:** ACB individuals are at high risk of having diabetes, requiring culturally tailored peer-based health promotion strategies to reduce diabetes prevalence and risk. HbA1c is a culturally acceptable and statistically more capable measure than OGTT in identifying individuals with prediabetes. Further longitudinal research is needed.

## 1. Introduction

Over the past decade, the occurrence of dysglycemia—which encompasses both diabetes and high blood sugar levels approaching those of diabetes (referred to as prediabetes)—has seen a 50% increase in Canada, and this trend is anticipated to persist [1,2,3]. Currently, one out of every three Canadians has some level of dysglycemia. The rate of diabetes has risen more significantly among younger adults (ages 20–49) than among older adults (ages 50 and above) both in Ontario and across Canada [1,3,4]. Young adults diagnosed with diabetes at an early age exhibit a distinct and more severe disease process compared to their older counterparts, requiring more immediate and intensive interventions to manage blood sugar levels [5,6].

Researchers report diabetes as colour-coded, meaning it disproportionately affects certain ethnocultural groups such as Blacks and South Asians. This group is vulnerable, even if they do not think they are (i.e., they are vulnerable but unaware). Evidence supports this report, as Black Canadians have faced a doubling of diabetes prevalence in the last decade [7]. Although there is substantial evidence of the increased risk of diabetes in the Black Canadian population, there is scanty research on the Black population aged 18–39 years. Diabetes Canada recommends universal glycated hemoglobin A1c (HbA1c) cut-offs for diagnosing diabetes; however, studies suggest ethnocultural adaptability for HbA1c as a diagnostic tool [8]. Current diabetes diagnoses rely on a cut-off of 6.5% [9,10]. The benefits of HbA1c for diabetes diagnosis are numerous, including the fact that there are no set time of day requirements and no special preparations. In addition to HbA1c tests, other tests, such as fasting plasma glucose (FPG) and oral glucose tolerance tests (OGTT), are used for diabetes diagnosis [3,10]. FPG requires long fasting hours, which is not always appreciated.

### Cultural Relativism and Diabetes

Even though culture and ethnicity are not interchangeable, for operational purposes, we used Black Afro-Canadian or Caribbean-Canadian ethnicity as a proxy for the culture of ACB individuals in this study to understand the trend and distribution of dysglycaemia in this demographic [11] and their behavioural patterns. Among young adults who face more severe disease progression, racial and ethnic minority populations face an even greater burden [12]. Over the past decade, the cases of diabetes among Black Canadians have seen a twofold increase, with data indicating that Black individuals are twice as likely to develop diabetes compared to other populations [7,13,14,15]. In Ontario, the self-reported diabetes prevalence is 8.5% in Black individuals, which is double the rate of 4.2% observed in White individuals [7]. Beyond the higher prevalence of diabetes, Black communities face higher rates of complications, including hypertension, retinopathy, and kidney disease [7]. Furthermore, previous scholarly work in this regard has suggested that immigrants of specific ethnic groups, such as South Asians, Chinese, and Blacks, develop diabetes and other metabolic disorders at a higher rate, at an earlier age, and within the lower ranges of body mass index (BMI) [16,17]. Despite this risk discrepancy between the Black minority and non-minority populations, current glycated hemoglobin A1c (HbA1c) cutoffs remain non-specific in evaluating the glycemic status of ABC communities [18]. Evidently, the increasing prevalence of diabetes in Black populations is the result of intersections between possible biological differences and access to economic, social, and cultural factors [19]. These intersections highlight that the adequate detection, evaluation, and mitigation of dysglycemia in the ABC population requires the incorporation of cultural relativity.

Incorporating cultural relativity in diabetes diagnoses allows for early and accurate identification and the subsequent mitigation of dysglycemia. This is because cultural characteristics influence illness perceptions and, thus, behaviour [11]. The early detection of diabetes risk using a culturally relative lens can enhance disease management and lower the chances of developing complications, thereby alleviating some of the disease’s adverse health effects [11,20]. Additionally, identifying risk factors and adopting healthier and culturally appropriate lifestyle habits can delay or even prevent the onset of diabetes in those identified with prediabetes [21], which is a critical public health demand. However, many people with prediabetes or in the initial phases of diabetes do not show symptoms, and this is exacerbated in Black communities, leading to delayed diagnosis and treatment [3,22,23].

The prevalence of dysglycemia is increasing in the ACB community. At the same time, culturally acceptable screening measures are not comprehensive and not tailored to address this population’s unique health challenges and lifestyle factors. The Public Health Agency of Canada (PHAC) and the Canadian Task Force of Preventive Health Care developed the Canadian Diabetes Risk Questionnaire (CANRISK) tool to screen diabetes in asymptomatic people. The CANRISK tool is adapted from other diabetes questionnaires that included predictive variables such as the ethnic origin of biological parents and questions related to family history. This tool is cost-effective and helpful in identifying high-risk individuals and prompting further tests [24]. Although widely accepted, some ethnic subgroups, like young Black Canadians, are underrepresented in this tool.

In that context, this study selected the research questions:What is the percentage of ACB residents (Toronto) aged 18–39 years old who have diabetes and prediabetes?What risks are identified in this population that influence blood sugar levels?Is HbA1c a more sensitive and culturally acceptable test than OGTT to diagnose diabetes in ACB communities?What are the recommendations for framing a culturally acceptable diabetes education protocol for ethnic populations?

## 2. Methods

This descriptive study collected data between June 2023 and September 2024 to measure the distribution of diabetes and prediabetes among ACB individuals aged 18–39 years in Toronto (Canada) and to identify the risks of prediabetes and diabetes among this population [25]. This study used a mixed-method approach (quantitative, qualitative, and blood work) to collect data [26]. The Public Health Research Ethics Board approved this study protocol for engaging with human participants.

### 2.1. Participant Recruitment

Participants included Black Toronto residents aged between 18 and 39 years who self-identified as of African and Caribbean ancestry. Study participants (*n* = 272) were recruited through community engagement, outreach, flyers, and peer-to-peer referrals using a purposive heterogeneous sampling technique to ensure a maximal variation of samples [27,28]. This technique allowed for the representation of broader ethnic and geographic diversity, including fourteen African countries consisting of five regions (East, West, Central, North, and South) and five Caribbean countries (Figure 1). The participants’ countries of origin were Côte d’Ivoire, Ghana, Nigeria, Chad, Ethiopia, Kenya, Rwanda, Tanzania, Uganda, South Sudan, Cameroon, Angola, South Africa, and Zimbabwe. Caribbean immigrants were from the islands of the Caribbean Sea (Bahamas, Barbados, Guyana, Jamaica, and St. Vincent) [29].

Participants were excluded if they had a previous diagnosis of diabetes with or without medication, if they were not in a physical state to travel to the centre, or if they were not in a mental state to understand or give informed consent, and if they were pregnant.

### 2.2. Data Collection

After screening for eligibility and receiving informed consent, the trained Peer Researcher (PR) recruited 272 participants. The PR (OM—a Black African immigrant physician) and a volunteer (SN—a Black African medical student) measured the participants’ height, weight, waist circumference, and blood pressure. The participants were guided to a standard blood testing laboratory in the same building (LifeLabs) for measuring HbA1c and for the oral glucose tolerance test (OGTT) following the standard protocol from the Public Health Agency of Canada (PHAC). When the participants had to wait for two hours for collection of the second blood sample for OGTT, the PR collected demographic, behavioural, and lifestyle data from the participants using the self-reported CANRISK tool.

### 2.3. Diagnoses Thresholds

Based on the Diabetes Canada Guideline, prediabetes was labelled for a fasting plasma glucose (FPG) of 6.1–6.9 mmol/L, or a plasma glucose level two hours after a glucose drink (2hPG) of 7.8–11.0 mmol/L, or glycated hemoglobin (HbA1c) of 6.0–6.4% [9]. The same guideline defined diabetes as an FPG of 7.0 mmol/L or higher, a 2hPG glucose level of 11.1 mmol/L or higher, or an HbA1c of 6.5% or higher [9].

### 2.4. Single-Subject Research Design

To capture more in-depth information (exploratory sequential approach) about any missing diabetes-related attributable behaviour of the ACB individuals, a single-subject research design was adopted to follow up on 41 (*n* = 41) randomly selected participants (out of the 272) from three subgroups: (i) at-risk group, (ii) prediabetes group, and (iii) diabetes group [30]. These qualitative data on influencing behavioural and social factors were analyzed following an interpretive phenomenological analysis protocol [31]. This longitudinal feature, which was subsequently added to the cross-sectional study design, added to the mixed-method nature of this research study [26].

### 2.5. Data Analysis

Quantitative data were analyzed using the International Machine Business Corporation Statistical Package for the Social Sciences (IBM SPSS) software 29.0.2.0 (20). ANOVA and regression analyses were performed to find any relationships between blood sugar and anthropometric, demographic, or behavioural variables. The qualitative data were analyzed manually using Braun and Clarke’s reflexive thematic approach [32]. The behavioural and social factor data were analyzed using an interpretive phenomenological analysis (IPA) protocol [31]. IPA explores the details of personal lived experiences and how people explain those experiences as their life events. The next step was triangulating quantitative and qualitative data to interpret the themes and relationships.

For qualitative data, participants in the at-risk, prediabetes, and diabetes groups (*n* = 41) were randomly selected for a qualitative interview to learn more about their behavioural, cultural, and familial attributes. The collected qualitative data were analyzed following the IPA approach and matched with the quantitative data collected by the CANRISK tool [31]. This data included participants’ lifestyle, diet, family history of diabetes, mental health, and changes in physical and lifestyle factors resulting from their migration to Canada to identify a trend within each group.

## 3. Results

### 3.1. Demographic and Geographical Distribution

The mean age of the participants (N = 272) was 30.8 ± 4.9 years (ranging between 18 and 39 years), including 38.2% of participants aged between 18 and 29 years and 61.8% between 30 and 39 years (Table 1). In this study, 65.8% of the participants self-identified as women, 34.2% as men, and none identified as non-binary (Table 1).

### 3.2. Glycated Haemoglobin (HbA1c) and Oral Glucose Tolerance Test (OGTT)

The mean HbA1c for these participants was 5.5% ± 0.5 (range 4.2–10.0%) (Table 2). Based on HbA1c levels, 1.5% (*n* = 4) participants were found to have diabetes, 9.2% (*n* = 25) had pre-diabetes, 44.9% (*n* = 122) were at risk for diabetes/pre-diabetes, and 44.5% (*n* = 121) were normal (Table 3).

The mean FPG value was 4.8 ± 0.6 mmol/L (range 3.5–10.1 mmol/L) (Table 2). This measure identified 1.5% (*n* = 4) participants as having diabetes, 2.6% (*n* = 7) participants as having pre-diabetes, and 96.0% (*n* = 261) participants as being normal or at risk for diabetes/pre-diabetes (Table 3).

The mean 2hPG after 75 g glucose intake was 5.7 ± 1.6 mmol/L (range 3.3–17.1 mmol/L) (Table 2). Based on 2hPG levels, 1.5% (*n* = 4) participants were found to have diabetes, 7.0% (*n* = 19) participants were found to have prediabetes, and 91.5% (*n* = 249) were found to be normal or at risk for diabetes/pre-diabetes (Table 3).

The mean height of the participants was 166.2 ± 8.7 cm (range 144.1–198.6 cm) (Table 2). The mean body weight was 78.0 ± 17.3 kg (range 37.5–146.2 kg) (Table 2). The mean BMI was 28.2 ± 5.8 kg/m^2^ (range 15.8–51.5 kg/m^2^) (Table 2). The mean waist circumference (WC) was 85.8 ± 13.1 cm (range 56.5–132.0 cm) (Table 2).

Table 3 indicates that HbA1c captured all of the cases of diabetes as identified by OGTT (FBS and 2hPG), more cases of prediabetes (9.2% by HbA1c vs. 2.6% by FBS and 7.0% by 2hPG) and more cases of at-risk individuals (44.9% by HbA1c vs. 3.7% by FBS). This means HbA1c has the same level of specificity as OGTT but a higher level of sensitivity to screen at-risk populations to prevent end-organ damage (micro and macrovascular consequences) by early diagnosis and prevention.

### 3.3. Identified Risk Factors

Figure 2 indicates a linear trend of the severity of glucose intolerance with increasing body mass index (BMI), weight, and waist circumference. For participants with diabetes, 50% of mothers and 25% of fathers had a history of having diabetes. In comparison, 16% of mothers of the participants with pre-diabetes and 24% of fathers had a history of having diabetes.

Participants with diabetes (*n* = 4) had a mean BMI of 37.2 ± 1.4 kg/m^2^, a WC of 102.7 ± 7.6 cm, and a weight of 97.7 ± 4.9 kg (Figure 2). One-fourth of them (25%) did not do intentional physical activity, while 75% did not eat fruits or vegetables daily. Participants with prediabetes (*n* = 25) had a mean BMI of 33.2 ± 5.9 kg/m^2^, a WC of 99.1 ± 10.6 cm, and a weight of 94.2 ± 16.2 kg (Figure 2). About 40% did not do intentional physical activity, while 72% did not eat fruits or vegetables daily.

Participants in the ‘At Risk’ category (*n* = 122, by HbA1c levels) had a mean BMI of 28.7 ± 5.5 kg/m^2^, a WC of 86.9 ± 12.6 cm, and a weight of 79.3 ± 17.4 kg (Figure 2). About 18% of them did not do any physical activity, and 79.5% did not eat any fruits or vegetables daily. About 10.7% of the participant’s mothers and 18.9% of fathers had a history of having diabetes.

Further, interpretive analysis showed a linear relationship between increasing blood sugar and the anthropometric measures of weight, BMI, and WC (F = 10.4, df1 = 3, df2 = 266, *p* < 0.001). Age was also proportionately related to having diabetes in this subset of the population (F = 3.5, df1 = 2, df2 = 269, *p* < 0.05). The participants’ mothers having diabetes was found to be more of a predictor (*p* = 0.1) of higher blood sugar levels than the fathers having diabetes (*p* = 0.4).

To understand the predictors of diabetes, a factor analysis was performed for participants who were in the ‘At Risk’ group (*n* = 122). The statistical acceptability of the analysis was tested by the Kaiser–Meyer–Olkin (KMO) measure of sampling adequacy, which was 0.735 (greater than 0.60), and Bartlett’s test of sphericity, which was significant (*p* < 0.01). This allowed the team to proceed with exploratory factor analysis, where WC, BMI, and weight were found to be the most important factors for increased blood sugar in these populations (the Eigenvalues were 0.953, 0.941, and 0.934, respectively).

### 3.4. Lifestyle, Diet, and Mental Health Challenges

In the ‘At Risk’ group, 60% of respondents agreed with consuming high-calorie food weekly. They also agreed that they ate when hungry or when food was available, and generally ate after 8 PM. About 20% of respondents reported a family history of diabetes in either one or both biological parents. Regarding lifestyle, all participants confirmed that they did not smoke or drink and agreed with only participating in slightly physical activities such as brisk walking. Likewise, 80% of participants expressed feelings of anxiety or depression that originated from either adjustment to the harsh weather in Canada and the Canadian workplace, difficulty in finding a job, or a combination of these.

## 4. Discussion

The definition of ‘Black African’ is agreed upon. However, defining ‘Black Caribbean’ required a literature search and peer consultation. Geographically, the Caribbean includes the countries surrounding the Caribbean Sea, including Venezuela, Colombia, Panama, Costa Rica, Nicaragua, Honduras, Guatemala, Belize, the Yucatan Peninsula of Mexico, Cuba, Hispaniola, Jamaica, Puerto Rico, the Virgin Islands, and Trinidad [33]. However, many people, including participants in this study, may describe themselves as Caribbean persons despite not being from a geographic Caribbean region. This discrepancy is in part due to colonial connotations, which have led to the term Caribbean being used to describe many geographic regions that have little to do with physical geography but instead arise from the cultural, economic, ethnic, and political similarities between countries of the Caribbean and those nearby [34]. Given these discrepancies, in the current study, the term Black Caribbean was used to characterize African-descendant residents of the islands in the Caribbean Sea, some South and Central American countries (Guyana, Suriname, and Belize), and islands in the Atlantic (Turks and Caicos Islands) [29]. The participants represented fourteen African countries in five regions and five Caribbean countries.

Culture and ethnicity are complex topics and not always interchangeable; however, the study team decided to use ethnicity data as a proxy indicator for culture data in this study. Other studies also adopted a similar consideration [11]. This study collected representative data from immigrants from many African and Caribbean countries (of different ethnicities) to share information on the various cultures across the continents.

The mean HbA1c of the participants (*n* = 272) was found to be 5.5 mmol/L (higher than the standard Canadian value), which fell into the category of ‘At Risk’ of diabetes as per Canadian guidelines. Taloyan et al. (2021) reported that the HbA1c of African immigrants was higher than Swedish-born participants [35]. Herman and Cohen (2012) also reported that Black people had a higher HbA1c than non-Blacks [8].

This study screened 1.4% of participants with diabetes, one of whom (with a HbA1c of 10 and a BMI of 28.26 kg/m^2^) required urgent primary care consultation. As a community-based action research study through an equity lens, participants and their communities who require urgent care receive immediate benefits from the research activity. This is Community-based Research’s social and research commitment [36]. Following the current diagnostic criteria from Diabetes Canada guidelines, HbA1c screened 47.4% of cases of prediabetes (Chi-square value 113.2, df 9, *p* < 0.001) and 75% of cases of diabetes (Chi-square value 199.5, df 6, *p* < 0.001) diagnosed by OGTT [9]. This study found HbA1c to be 24% more sensitive than OGTT for screening prediabetes. Other studies with Black and other ethnic minorities also observed that HbA1c’s diagnostic sensitivity was higher [37,38]. This observation recommends that HbA1c can be used as a sensitive screening tool for ethnic minorities.

To test the sensitivity and specificity of HbA1c for participants diagnosed by OGTT with prediabetes and diabetes, the receiver operator characteristic (ROC) curve was attempted. The area value of the analyzed prediabetes data was 0.881, *p* < 0.001, and CI 0.817–0.944, meaning it was a good fit for analysis. The HbA1c cut-off value of HbA1c for prediabetes was 5.7% (Figure 3).

In comparison, the cut-off value for diabetes was found to be 6.4% (area value 0.969, *p* < 0.001, CI 0.969–1.007) (the poor sample size made the interpretation invalid for the diabetes group). The standard HbA1c value for prediabetes is 6.0–6.4%, and for diabetes, 6.5% or more [9]. This analysis generated a hypothesis that setting a lower HbA1c cut-off would be best for diagnosing prediabetes and diabetes in ACB individuals. This can be implemented after a study with larger sample size and geographically wider area is performed. If the hypothesis is proven correct, the Canadian Guidelines should be revised, raising the cut-off values. This finding can be tested with other ethnic groups, such as South Asians or Indigenous communities, to determine a culturally relative, not a universal, cut-off value for diabetes.

As a test procedure, the precision and compliance with the OGTT were lower than HbA1c. This was identified in qualitative discussions with participants whose fasting blood sugar was more than the blood sugar level measured 2 h after a 75 g glucose drink. Their food behaviour of having supper late in the evening and their cultural preferences for a more carbohydrate-rich diet caused a higher fasting sugar level. They did not drink the whole 75 g glucose because this was “too sweet to drink in the morning”. They did not trust what was free to drink and poured it out in the bathroom. Some saved it for their children at home and drank water from their bottles. Some drank it slowly over an hour, mixed with water. After this information, we drew attention to the LifeLabs quality assurance team. They trained their staff to check and control the process so that everyone finished the glucose in front of them within five minutes. We also tried to control the glucose drinking behaviour of the study participants during the test, but these were the feelings of these ACB individuals, as they mentioned. Therefore, we recommend using HbA1c and random blood sugar testing for diagnostic purposes for ACBs and also for other ethnic minorities if they have similar cultural traits. This will also reduce costs and increase patient compliance without compromising diagnostic precision.

Qualitative data also explored how food insecurity affected their food behaviour and the cultural traits of preferring carbohydrate-rich food and eating late in the evening. The participants considered the at-risk nature of these findings primarily due to their diet, lifestyle, and immigration status rather than because of family history. Their mental health had been an issue that needed to be addressed. They had settlement anxiety and workplace culture anxiety. Some of them drank alcohol as a habit, while most of them did not.

Considering previous study findings that diabetes affects some ethnicities or racial groups more than others, ethnic-based or culture-based intervention is required [39,40,41,42]. This study emphasizes that immigrants in early age groups must be screened with the updated CANRISK tool [43].

### Strengths and Limitations

Participants were recruited from multiple Access Alliance locations across Toronto. Despite a meticulous design to collect representative samples of the target population and rigorous data analysis techniques, the sample size was small, needed mindful interpretation, and a larger study for generalization. Collecting qualitative data to interpret quantitative findings was a supplement to ensure data accuracy and validity. However, culturally competent further longitudinal research on this issue with a larger sample size is required to generalize the findings. Future studies must consider lifestyle and behavioural indicators more meticulously at the research design level.

## 5. Conclusions

This screen identified that ACB individuals aged 18–39 years are vulnerable to diabetes and prediabetes, with over 44% of them being at risk. Their ethnicity is a factor; however, their food habit, lifestyle, and settlement challenges in Canada are other contributory factors for dysglycemia. The participants had unique food cultures and preferences; therefore, they will need culturally appropriate diet education from their peers (proximity principle) and flexibility in dietary options. As such, proximity and flexibility can be very instrumental tools for successful diabetes education programs. The intersectional analysis found HbA1c to be a more sensitive and culturally acceptable measure than OGTT to screen ACB individuals into at risk/prediabetes groups. HbA1c level was found to be higher in ACB populations, and which requires larger longitudinal research to set a culture-relative equity-focused cut-off for ethnic minorities (instead of the current universal cut-off) to diagnose prediabetes and diabetes. Considering mental health as a significant issue in ethnic minorities with dysglycemia, diabetes education will also require incorporating mental health consultations to address this.

## Figures and Tables

**Figure 1 ijerph-22-00085-f001:**
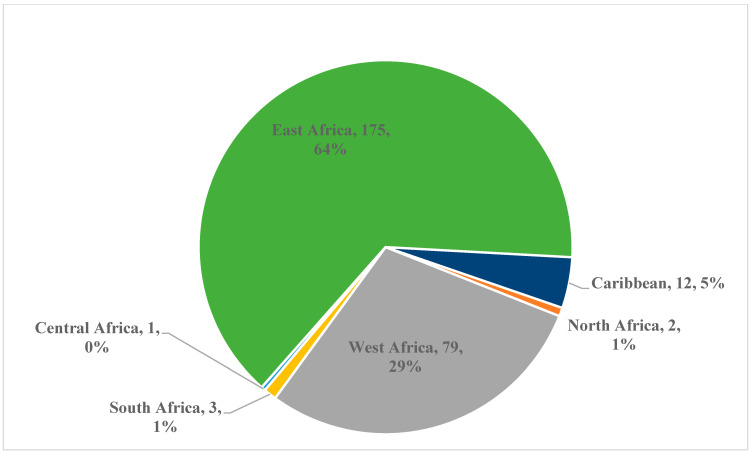
Geographical distribution of participants (*n* = 272) (from 14 African and 5 Caribbean countries).

**Figure 2 ijerph-22-00085-f002:**
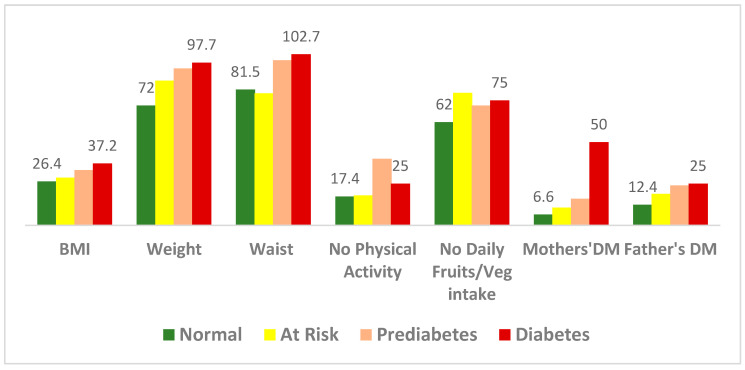
Relation of risk factors affecting participants with normal levels of blood glucose, at Risk, prediabetes, and diabetes (*n* = 272).

**Figure 3 ijerph-22-00085-f003:**
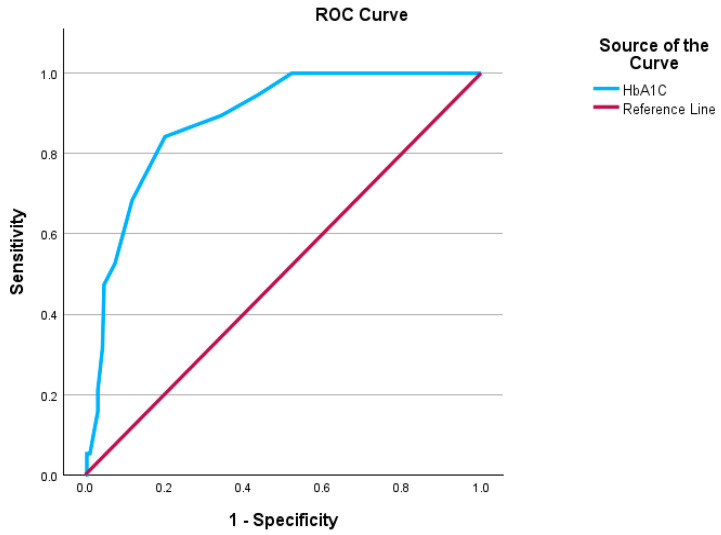
ROC Curve of HbA1c for diagnosed prediabetes by 2hPG (*n* = 19).

**Table 1 ijerph-22-00085-t001:** Characteristics of study participants (*n* = 272).

	Number	Percentage (%)
**Gender**		
Male	93	34.2
Female	179	65.8
**Age**		
Mean 30.8 (±4.9)		
Age Group 1 (18–29 years)	104	38.2
Age Group 2 (30–39 years)	168	61.8
**Geographical Region of Origin**		
(14 African + 5 Caribbean countries)		
North Africa	2	0.7
West Africa	79	29.1
Central Africa	1	0.4
East Africa	175	64.3
South Africa	3	1.1
Caribbean	12	4.4
**Health Insurance Status**		
Insured (OHIP)	50	18
Non-Insured	222	82
**Level of Education**		
Some High School or Less	15	5.5
High School Diploma	41	15.1
Some College/University	42	15.4
College/University Degree	174	64

**Table 2 ijerph-22-00085-t002:** Participants’ anthropometric and blood reports (*n* = 272).

	Mean Value ± SD	Range (Min–Max)
Height (cm)	166.2 ± 8.7	144.1–198.6
Body weight (kg)	78.0 ± 17.3	37.5–146.2
BMI (kg/m^2^)	28.2 ± 5.8	15.8–51.2
Waist Circumference (cm)	85.8 ± 13.1	56.5–132.0
FBS (mmol/L)	4.8 ± 0.6	3.5–10.1
2hPG (mmol/L)	5.7 ± 1.6	3.3–17.1
HbA1c (%)	5.5 ± 0.5	4.2–10.0

**Table 3 ijerph-22-00085-t003:** Distribution of participants by blood sugar levels (*n* = 272).

	By HbA1c	By FBS	By 2hPG
Normal	121 (44.5%)	251 (92.3%)	249 (91.5%)
At Risk	122 (44.9%)	10 (3.7%)	
Prediabetes	25 (9.2%)	7 (2.6%)	19 (7.0%)
Diabetes	4 (1.5%)	4 (1.5%)	4 (1.5%)

## Data Availability

Data cannot be shared publicly because of Personal Health Information confidentiality and data security protocol. All data is saved for five years in the encrypted computer of Access Alliance and is available on request with the condition of anonymity.

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
