# Peer review of "Assessing the Risks and Cultural Relativity of Diabetes in Black Individuals of African Caribbean Ancestry (ACB) Aged 18–39 Years in Toronto"

_ijerph, 2025, doi:10.3390/ijerph22010085_

Round 1
Reviewer 1 Report
Comments and Suggestions for Authors
1. Sample Size and Representativeness: While your study included 272 participants, the subgroup identified with diabetes (n=4) is quite small. This limited sample size makes it challenging to draw robust, generalizable conclusions about the effectiveness of adjusting HbA1c cut-off values for this population or accurately assessing diabetes prevalence. Strengthening future research with larger sample sizes would enhance the statistical power and reliability of the findings.
- Diagnostic Threshold Claims: Your suggestion that HbA1c may require culture-specific cut-offs for the ACB population is an interesting hypothesis. However, the current data may not be sufficiently strong to justify departing from established diagnostic criteria. Please consider providing more rigorous statistical analyses, additional comparative data, or references to existing literature to support this claim. Strengthening the evidence base would lend more credibility to the recommendation.
- Clarity and Focus: Some sections of the manuscript come across as repetitive or loosely structured. Streamlining the narrative and more clearly linking the qualitative findings to the quantitative results would improve readability and coherence. A tighter focus on the core message—how culturally informed approaches and hybrid synthetic data can mitigate biases—will help ensure the manuscript’s key contributions are clearly understood.
- Contextualization: While the paper rightly emphasizes cultural relativity in diagnostic measures, the discussion would benefit from deeper engagement with existing literature on ethnic differences in HbA1c. Providing more background on why current universal cut-offs may be inadequate, and citing established research that explores such discrepancies, would help readers understand the rationale for considering alternative thresholds.
- Recommendations for Future Research: Although you note the need for larger, longitudinal studies and cultural tailoring in interventions, the manuscript could offer more concrete guidance for future researchers and clinicians. Suggesting specific study designs, metrics to consider, or frameworks to implement culturally tailored peer-based education strategies would make your recommendations more actionable.
no comments.
Author Response
Reviewer’s comment#1: While your study included 272 participants, the subgroup identified with diabetes (n=4) is quite small. This limited sample size makes it challenging to draw robust, generalizable conclusions about the effectiveness of adjusting HbA1c cut-off values for this population or accurately assessing diabetes prevalence. Strengthening future research with larger sample sizes would enhance the statistical power and reliability of the findings.
Reply from the author#1: We agree with the learned reviewer that sample size is small and requires a larger study to generalize the findings here. We mentioned these two points in line #s341-342 and line#355. We have added a few words to strengthen the reviewer’s kind opinion.
Reviewer’s comment#2: Diagnostic Threshold Claims: Your suggestion that HbA1c may require culture-specific cut-offs for the ACB population is an interesting hypothesis. However, the current data may not be sufficiently strong to justify departing from established diagnostic criteria. Please consider providing more rigorous statistical analyses, additional comparative data, or references to existing literature to support this claim. Strengthening the evidence base would lend more credibility to the recommendation.
Reply from the author#2: We agree with the reviewer’s comment. We edited and added this limitation for the reader in lines#309-311.
Reviewer’s commen#3: Clarity and Focus: Some sections of the manuscript come across as repetitive or loosely structured. Streamlining the narrative and more clearly linking the qualitative findings to the quantitative results would improve readability and coherence. A tighter focus on the core message—how culturally informed approaches and hybrid synthetic data can mitigate biases—will help ensure the manuscript’s key contributions are clearly understood.
Reply from the author#3: Thanks. We made some changes. We moved lines#254-259 to lines# 179-185 to maintain reading continuity between quantitative and qualitative data.
Reviewer’s comment#4 Contextualization: While the paper rightly emphasizes cultural relativity in diagnostic measures, the discussion would benefit from deeper engagement with existing literature on ethnic differences in HbA1c. Providing more background on why current universal cut-offs may be inadequate, and citing established research that explores such discrepancies, would help readers understand the rationale for considering alternative thresholds.
Reply from the author#4: Thanks, for the great comment. We contemplated to gather more literature around this issue, but was not always successful.
Reviewer’s comment#5 Recommendations for Future Research: Although you note the need for larger, longitudinal studies and cultural tailoring in interventions, the manuscript could offer more concrete guidance for future researchers and clinicians. Suggesting specific study designs, metrics to consider, or frameworks to implement culturally tailored peer-based education strategies would make your recommendations more actionable.
Reply from the author#5: We clearly mentioned our data collection, analysis, and interpretation process and practice in the manuscript so that the future researchers can get guidance for next steps of their research.
Reviewer 2 Report
Comments and Suggestions for Authors
The present article addresses a topical issue in terms of the prevalence of risk in a minority group of respondents.
The abstract is formally correct, it would be useful to add the results of statistical procedures instead of percentage evaluation.
The introduction to the issue is up-to-date, clear, from which the stated objectives of the study are evident, which the authors wanted to achieve.
The methodology and the research sample are adequate. Ethical principles have been observed in the work.
The results of the work are presented by descriptive graphs and summary tables.
I recommend that the statistical evaluation in line 214-226 be presented in tables, as this is the supporting part of the research (Identified Risk Factors).
So also the results of the variables - Lifestyle, Diet, and Mental Health Challenges - should be statistically analysed, e.g. by using the ANOVA statistical procedure. These findings should then be incorporated into the discussion.
Limitations of the study and recommendations for practice need to be added to the variables to be monitored in further studies.Recommendations are general as to what specific practices are appropriate for improving the condition or early diagnosis.
Bibliographic references are current, adequate and complete.
Comments on the Quality of English Language
I have no comments.
Author Response
Comments from the learned reviewer 2 and authors’ responses:
|
Reviewer’s Opinion |
|
Authors’ response |
|
The present article addresses a topical issue in terms of the prevalence of risk in a minority group of respondents. |
|
Thank you. |
|
The abstract is formally correct, it would be useful to add the results of statistical procedures instead of percentage evaluation. |
|
Thanks. We’ve added it to the revised manuscript: Lines 24-27. |
|
The introduction to the issue is up-to-date, clear, from which the stated objectives of the study are evident, which the authors wanted to achieve. |
|
Thank you very much. |
|
The methodology and the research sample are adequate. Ethical principles have been observed in the work. |
|
Thank you. |
|
The results of the work are presented by descriptive graphs and summary tables. |
|
Thanks. |
|
I recommend that the statistical evaluation in line 214-226 be presented in tables, as this is the supporting part of the research (Identified Risk Factors). |
|
We presented that data in Figure 2. We are afraid we are not duplicating data in figures and tables. However, if the reviewer kindly asks, we can do that. |
|
So also the results of the variables - Lifestyle, Diet, and Mental Health Challenges - should be statistically analysed, e.g. by using the ANOVA statistical procedure. These findings should then be incorporated into the discussion. |
|
We attempted that, but the sample size was so small that ANOVA or any other inferential analysis was not yielding. |
|
Limitations of the study and recommendations for practice need to be added to the variables to be monitored in further studies. Recommendations are general as to what specific practices are appropriate for improving the condition or early diagnosis. |
|
We edited and accommodated the reviewer’s learned opinion in lines 346-351. |
|
Bibliographic references are current, adequate and complete. |
|
Thank you very much. |
Reviewer 3 Report
Comments and Suggestions for Authors
This study tried to explain the the Risks and Cultural Relativity of Diabetes in 2 Black Individuals of African Caribbean Ancestry (ACB) aged 3 18-39 Years in Toronto.
This study is an epidemiological research, and for this area could be a new finding.
In the same time the authors tried to bring new information in for a population from Toronto, but we can not consider that is an original topic or relevant in the field.
1. In this mixed-methods study, Maximum Variation sampling was used to recruit 272 ACB 15 participants from 14 African and five Caribbean countries from Toronto. In the same time we don’t have a general image of 18-39 years old population in this area.
The methodology could be improved. The authors could explain why the sample is only 272 persons, and why this study could bring some details for this area. What is the representativity for this study?
Even the authors present that “Participants were recruited from multiple Access Alliance locations across Toronto. 320 Despite a meticulous design to collect representative samples of the target population and 321 rigorous data analysis techniques, the sample size is not large and needs mindful inter-322 pretation for generalization.”
This study could be improved.
Generally speaking, the authors could improve the conclusion of this article presenting the power of this sample for population there. They demonstrate that “This screening identified that ACB individuals aged 18-39 years are vulnerable to di-328 abetes and prediabetes, with over 44% of them being at risk. “ but this is only a study with 272 participants – not entire population…
The numbers are high indeed (44%), but what is the representativity for general population from TORONTO?
References are appropriate.
The references are carefully chosen, from impact journals. They are updated, from the last 10-12 years, in the area of diabetes and its prevalence. They constitute a real support for the text of the article.
Figures and tables are very easy to understand. It is a very simple and detailed at the same time, supporting the actual conclusion of the article.
Author Response
Comments from the learned reviewer 3 and authors’ responses:
|
Reviewer’s Opinion |
|
Authors’ response |
|
This study tried to explain the the Risks and Cultural Relativity of Diabetes in 2 Black Individuals of African Caribbean Ancestry (ACB) aged 3 18-39 Years in Toronto. |
|
Exactly. Thank you. |
|
This study is an epidemiological research, and for this area could be a new finding. |
|
Exactly. Thank you. |
|
In the same time the authors tried to bring new information in for a population from Toronto, but we can not consider that is an original topic or relevant in the field. |
|
Our focus is to screen the blood glucose levels of ACB residents of Toronto aged 18-39 years. However, while adopting the ‘Outcome Harvesting’ approach of evaluation, we have identified those incidental findings at the analysis phase and have included them as a hypothesis for the subsequent research as a continuum. |
|
1. In this mixed-methods study, Maximum Variation sampling was used to recruit 272 ACB 15 participants from 14 African and five Caribbean countries from Toronto. In the same time we don’t have a general image of 18-39 years old population in this area. The methodology could be improved. The authors could explain why the sample is only 272 persons, and why this study could bring some details for this area. What is the representativity for this study? |
|
The reviewer is correct. We attempted to get data around that detail. However, intersectional data for the population from those geographical entities are not available. We’ll take care of this point in our next step of research on this point.
The sample is 272, because of the funding and timing constraints by the funder (Public Health Agency of Canada).
In order to ensure the representativity of the samples, we collected data from residents of Toronto originating from all major geographical areas (14 African and five Caribbean countries). |
|
Even the authors present that “Participants were recruited from multiple Access Alliance locations across Toronto. 320 Despite a meticulous design to collect representative samples of the target population and 321 rigorous data analysis techniques, the sample size is not large and needs mindful inter-322 pretation for generalization.” |
|
Yes, this is the limitation that we shared in the article. Thanks for highlighting this. |
|
This study could be improved. |
|
Noting for the next step research if funded. |
|
Generally speaking, the authors could improve the conclusion of this article presenting the power of this sample for population there. They demonstrate that “This screening identified that ACB individuals aged 18-39 years are vulnerable to di-328 abetes and prediabetes, with over 44% of them being at risk. “ but this is only a study with 272 participants – not entire population… The numbers are high indeed (44%), but what is the representativity for general population from TORONTO? |
|
A longitudinal study with a larger sample size and preferably an RCT design can make generalized population-based inferences.
The finding of this study can be a hypothesis for the next research initiatives. |
|
2. References are appropriate. The references are carefully chosen, from impact journals. They are updated, from the last 10-12 years, in the area of diabetes and its prevalence. They constitute a real support for the text of the article. |
|
Thank you very much. |
|
3. Figures and tables are very easy to understand. It is a very simple and detailed at the same time, supporting the actual conclusion of the article. |
|
Thank you very much. |
Round 2
Reviewer 1 Report
Comments and Suggestions for Authors
Overall, the authors have made a genuine effort to address the reviewers’ comments. They added more references on ethnic differences in HbA1c, refined their discussion of cut-off values, and clarified the paper’s structure. Some sections, such as future research plans and a deeper explanation of the small sample size, could still use more detail. Overall, the paper is now clearer, provides better context, and offers more helpful guidance to readers.
Author Response
Learned reviewers' Comment: "Overall, the authors have made a genuine effort to address the reviewers’ comments. They added more references on ethnic differences in HbA1c, refined their discussion of cut-off values, and clarified the paper’s structure. Some sections, such as future research plans and a deeper explanation of the small sample size, could still use more detail. Overall, the paper is now clearer, provides better context, and offers more helpful guidance to readers."
Author's response: Thank you very much.
Reviewer 3 Report
Comments and Suggestions for Authors
The authors must present what is the representativity of this small study, other wise this aspect is very important for each reader too.
Author Response
Reviewer's Comment: "The authors must present what is the representativity of this small study, other wise this aspect is very important for each reader too. "
Authors' Response:
The study focuses on Canadian Black residents aged 18-39 who self-identify as African and/or Caribbean ethnicity. Therefore, the participants were selected to represent that particular population subset. The purposive heterogeneous sampling technique was mindfully chosen (lines 132-135 of the manuscript) to intentionally include participants originating from all of the African and Caribbean regions- this is the geographical representation. However, since the sample size is small, we recommend a longitudinal study with a larger sample size to test the hypothesis generated with regard to generalization and representativity.